# Effect of Mycolic Acids on Host Immunity and Lipid Metabolism

**DOI:** 10.3390/ijms25010396

**Published:** 2023-12-28

**Authors:** Haoran Wang, Dingpu Liu, Xiangmei Zhou

**Affiliations:** 1College of Veterinary Medicine, China Agricultural University, Beijing 100086, China; haoranwang0609@163.com (H.W.); ldingpu@126.com (D.L.); 2National Key Laboratory of Veterinary Public Health and Safety, Beijing 100086, China

**Keywords:** *Mycobacterium tuberculosis*, mycolic acids, immune response, lipid metabolism

## Abstract

Mycolic acids constitute pivotal constituents within the cell wall structure of *Mycobacterium tuberculosis*. Due to their structural diversity, the composition of mycolic acids exhibits substantial variations among different strains, endowing them with the distinctive label of being the ‘signature’ feature of mycobacterial species. Within *Mycobacterium tuberculosis*, the primary classes of mycolic acids include α-, keto-, and methoxy-mycolic acids. While these mycolic acids are predominantly esterified to the cell wall components (such as arabinogalactan, alginate, or glucose) of *Mycobacterium tuberculosis*, a fraction of free mycolic acids are secreted during in vitro growth of the bacterium. Remarkably, different types of mycolic acids possess varying capabilities to induce foamy macro-phages and trigger immune responses. Additionally, mycolic acids play a regulatory role in the lipid metabolism of host cells, thereby exerting influence over the progression of tuberculosis. Consequently, the multifaceted properties of mycolic acids shape the immune evasion strategy employed by *Mycobacterium tuberculosis*. A comprehensive understanding of mycolic acids is of paramount significance in the pursuit of developing tuberculosis therapeutics and unraveling the intricacies of its pathogenic mechanisms.

## 1. Introduction

Tuberculosis (TB) has been an ancient scourge for mankind for thousands of years and still poses a great threat to human health nowadays. According to the global tuberculosis report 2022, the number of TB deaths worldwide is estimated to have increased between 2019 and 2021, reversing the downward trend seen between 2005 and 2019 [1]. There were 1.6 million TB deaths in 2021, including 187,000 cases of HIV infection [2]. In particular, eight countries—India, Indonesia, China, the Philippines, Pakistan, Nigeria, Bangladesh, and the Democratic Republic of Congo—together account for more than 2/3 of the global TB burden [3]. *Mycobacterium tuberculosis* (*Mtb*) has evolved a range of immune evasion strategies as a successful pathogen, which is one of the reasons why TB is so difficult to eradicate. Mycolic acids (MAs) were first identified in *Mtb* by Stodola et al. in 1938, and are widely found in actinomycetes [4]. It is an important component of the cell wall of *Mtb* and plays a key role in stabilizing the cell wall structure, maintaining its dense nature and pathogenicity, and in antibiotics resistance [5]. Recent research reported that MAs also play a role in immune regulation. Thus, the modulatory effect of MAs on the body’s immunity seems to be a new direction in developing treatments of tuberculosis.

In this review, we focus on the regulatory role of MAs on organism immunity and lipid metabolism, to lay the foundation for unraveling the mechanisms of mycobacterial evasion.

## 2. Results

### 2.1. The Synthesis of Mycolic Acid

*Mtb* is a complex process involving multiple enzymatic steps and is crucial for the structure and function of the cell wall, which is a key factor in *Mtb*’s pathogenicity. The synthesis of mycolic acid is divided into three main parts. Firstly, the synthesis of mycolic acids begins with the formation of precursor molecules. The primary precursors are fatty acids, typically palmitic acid (C16:0), and stearic acid (C18:0), which are synthesized in the cytoplasm through fatty acid biosynthesis pathways [6]. Following synthesis, fatty acids are activated through coenzyme A (CoA) conjugation, forming acyl-CoA intermediates. These activated fatty acids are then transported to the cell membrane, the site of mycolic acid synthesis. MAs synthesis consists of two main parts, which are named fatty acid synthase I (FAS-Ⅰ) and fatty acid synthase Ⅱ (FAS-Ⅱ). The FAS-Ⅰ system includes multiple enzymes that iteratively add two-carbon units to the growing fatty acid chain. FAS-II further modifies the growing fatty acid chains. It is responsible for the introduction of cis double bonds, cyclopropane rings, and keto groups into the MAs [7].

Mature MAs in mycobacteria are transported to the cell wall under the regulation of various genes, including CmrA, MmpL3, and others [8,9]. In *Mtb*, MAs can exist in different forms, either esterified to the cell wall or in a free state, meaning MAs are covalently attached to the cell wall through ester bonds. Specifically, the hydroxyl group in the arabinogalactan is vertically linked to MAs via an ester bond formed with the carboxyl group in the MA molecule. This arabinogalactan is, in turn, attached to the peptidoglycan, which provides the essential structural framework for *Mtb* [10]. In addition to their covalent attachment, MAs can also be present in a non-bound, solvent-extractable form. These include trehalose monomycolate (TMM), trehalose dimycolate (TDM), glucose monomycolate (GMM), and glycerol monomycolate (GroMM) (Figure 1). These free forms of MAs have various roles in the mycobacterial cell envelope and interactions with the host environment.

In fact, in the process of MAs synthesis, the introduction of double bonds in some meromycolic chains or the insertion of some cyclopropyl groups and methyl groups constitute the structural diversity of MAs [11]. MAs have three mains’ structures types: α-MAs, keto-MAs, and methoxy-MAs [12]. α-MAs molecule has two cis-cyclopropane ring structures, which are the most abundant of the three mycolic acids (>50%), and the keto and methoxy-mycocerosic acid molecules have one cis- and one trans-cyclopropane ring structure, which are relatively less abundant (10–15%) [13] (Figure 2). The keto-MAs and methoxy-MAs are called oxidized mycolic acids due to the presence of functional groups containing oxygen.

The levels of these different species of MAs vary widely among *mycobacterium* species and strains, and numerous researchers have suggested that these differences in MA content appear to be directly related to virulence. For instance, in non-tuberculous mycobacteria such as *Mycobacterium smegmatis* (*M. semeg*), there is an absence of keto- and methoxy-type mycolic acids, which are instead replaced by α- and α’-mycolic acid types. Despite the subtle difference involving the replacement of the cyclopropane moiety in the molecular structure by a double bond or epoxypropane, the reduced permeability of the cell wall in the absence of keto- and methoxy-type mycolic acids in mycobacteria leads to alterations in the utilization of specific nutritional components during growth, resulting in reduced virulence [14]. Furthermore, the mycolic acid types in *Mtb* undergo changes in response to various factors, including environmental conditions and host interactions. For instance, long-term passaging of the *Mycobacterium* bovis BCG Pasteur through multiple generations has been associated with the loss of methoxy-type mycolic acids, which may also contribute to the observed reduction in virulence [15]. The researchers were able to determine the origin of the newly isolated *Mycobacterium* by comparing the MAs content. Therefore, MAs are also known as the “fingerprint” of *Mtb*.

Actually, the synthesis of MAs involves a multitude of enzymes, suggesting a promising avenue for the development of anti-tuberculosis medications. For instance, acyl carrier protein (AcpM), functioning as acyl-AcpM, assumes a central role within the FAS-II system. Wang observed that *M. semegs’* protein MSMEG_5634 is highly homologous to AcpM. Knockdown of MSMEG_5634 significantly reduced the sensitivity of the strain to drugs targeting FAS-II (e.g., isoniazid, triclosan), while high expression of MSMEG_5634 promoted the inhibitory effect of FAS-II inhibitors on mycobacterial growth [16].

Next, we introduce the influence of the structure and existence form of MAs on the immune regulation mechanism of the body in detail.

### 2.2. Immunomodulatory Effects of Glycosylated Mycolic Acids of Mycobacteria

Trehalose mycolates and related sugar mycolates are major glycolipids on the cell wall of *Mtb*, which exert immunological effects. Indeed, different sugar mycolates affect the host’s innate or adaptive immunity in different ways. In the context of this review, we summarized abundant evidence proving that glycosylated mycolic acids play an essential role in the invasion of *Mtb* into the host (Figure 3).

#### 2.2.1. Arabinose Monomycolates (AraMM)

Arabinose monomycolates (AraMMs) represent the predominant form of mycolic acids (MAs) within *Mycobacterium tuberculosis* (Mtb). These molecules serve a dual purpose by contributing to the structural integrity of Mtb and exerting immunological effects. Specifically, the complex mixture known as arabino-mycolate ester, comprising mono-AraMMs, tetra-arabinose tetra-mycolates, hexa-arabinose tetra-mycolates, and penta-arabinose tetra-mycolates, has been identified as an inducer of tumor necrosis factor-alpha (TNF-α) production. This induction occurs through mechanisms reliant on the Toll-like receptor 2 (TLR-2)/MyD88 pathway [17]. It is noteworthy that while AraMMs do elicit an inflammatory response, this response is comparatively modest in intensity when juxtaposed with the effects induced by other forms of MAs. Consequently, AraMMs are not considered to play a major role in immunomodulation.

#### 2.2.2. Trehalose Mycolates

Trehalose mycolates, whose components include trehalose dimycolate (TDM), trehalose monomycolate (TMM), and related glycoconates, such as trehalose dibehenate (TDB), are involved in a number of very strong immunological effects and are able to stimulate innate immunity or adaptive immunity.

The immune mechanism of TDM depends on C-type lectin receptors (CLR) [18]. CLRs are mainly expressed on the surface of the cell membrane of myeloid cells, monocytes, macrophages, and dendritic cells and consist of an extracellular segment containing a calcium-dependent glycan recognition region, a stalk region, a transmembrane region, and a short intracellular tail [19]. CLRs include both secretory and transmembrane types. The secretory CLRs are mainly represented by the collagen agglutinin family, and the transmembrane CLRs are divided into type I and type II according to their N-terminal orientation. Most CLRs are type II transmembrane proteins [20]. The main families of CLRs associated with MAs are macrophage-inducible C-type lectin receptors (Mincle) and macrophage C-type lectin (MCL). Mincle (also known as clec4e) was originally cloned as a transcriptional target of C/EBPβ [21]. Researchers found that Mincle is an activating receptor that couples with the Fc receptor (FcRγ) and mediates immune response [22]. We hold the opinion that Mincle stimulated with the TDM revealed increased NLRP3 inflammasome activation and downstream IL-1β cytokine release [23]. The NLRP3 inflammasome is a multiprotein complex consisting of NLRP3 with an additional adaptor molecule, an apoptosis-associated speck-like protein with a caspase recruitment domain encoded by Pycard (ASC), and caspase-1 [24]. Meanwhile, Mincle acts as a prototypical activating CLR after recognition of TDM. Through the full ITAM of the FcRγ chain adaptor, Mincle couples to splenic tyrosine kinase (Syk), which induces the CARD9/Bcl10/MALT1 complex activation-mediated NF-κB or MAPK signaling pathway to produce several cytokines, including TNF-α, IL-6, and IL-1β [25,26]. (Figure 3A).

However, Mincle is barely detectable in resting cells. This means that the massive activation of Mincle requires the involvement of other substances. It has been recently proposed that MCL is an FcRγ-coupled activating receptor that recognizes TDM and drives inducible Mincle expression upon stimulation by TDM. Further research identified that TDM failed to induce innate and acquired immunity in MCL-deficient mice [27] (Figure 3). Indeed, MCL and Mincle are co-regulated and depend on each other for their mutual surface expression [28,29]. Hence, Mincle and MCL dimerize would benefit phagocytosis, and Syk-mediated inflammatory responses [26]. Through phagocytosis by murine macrophages, TDM was found to colocalize with acidic phagosomes, implying that TDM can also inhibit the fusion of phagosomes with lysosomes, which, in turn, promotes intracellular survival of bacteria [30,31]. (Figure 3A). Thus, restricting the synthesis of TDM becomes an important way to limit *Mtb* infection.

Mincle recognition of TDM is not only limited to macrophages but also in neutrophils and it plays an important role in early lung inflammation [32]. In fact, neutrophils express higher levels of Mincle than macrophages and can respond immediately to TDM. For example, in response to TDM stimulation, Mincle on neutrophils synergistically activates the downstream MEK/ERK signaling pathway resulting in the release of pro-inflammatory factors. In addition, unlike macrophages, the FcRγ–Syk pathway on neutrophils recognized by TDM activates the expression of CD11b/CD18 on the cell surface, which, in turn, promotes neutrophil adhesion [33]. If the TLR2 receptor is activated by other structures of the *Mtb* cell wall at this time, it will enhance the expression of this pathway (Figure 3B).

Mincle activation on neutrophils is an important mode of host innate immune resistance to *Mtb* invasion, and the release of large amounts of inflammatory factors and the accumulation of leukocytes are positive in the early stages of the disease to control its progression. However, an excessive inflammatory response is highly threatening to the tissue structure of the lung. Therefore, the inflammatory response promoted by TDM needs to be taken into account in order to provide a theoretical basis for the treatment of tuberculosis.

*Mtb* is known to display a form of organized growth referred to as cording. The ability to cord has been associated with the virulence of *Mtb*, and TDM has been identified to be the cord factor [34]. Actually, MA is the key factor that makes trehalose mycolates virulent. Changing the structure of MAs can greatly affect the virulent effects of trehalose mycolates. It is reported that mutations in the *pcaA* gene affect the synthesis of an α-MA, methoxy-MA [35]. TDM purified from *pcaA* gene-deficient *Mtb* was hypo-inflammatory for macrophages and induced milder granulomatous inflammation in mice [36]. In addition, the *mmaA4* (methoxymycolate synthase 4) gene of *Mtb* functions to encode a methyl transferase that introduces an oxygen-containing modification of cell wall MAs. The *mmaA4*-inactivated mutants induce more IL-12p40 from mouse macrophages. IL-12p40 has the ability to enhance protective immunity in mice and humans infected with *Mtb* [37,38,39]. Nevertheless, the *Mtb* cyclopropane–mycolic acid synthase 2 (*cmaA2*) null mutant (Δ*cmaA2*), which lacks the trans-cyclopropanation of MAs, exhibits distinct outcomes. The absence of trans-cyclopropanation amplifies macrophage inflammatory responses induced by *Mtb*, thereby enhancing resistance against *Mtb* infection [40]. These findings underscore that the composition of MAs within TDM plays a direct role in determining the host cell’s immune response.

In conclusion, MAs in TDM are key components in eliciting host immune responses, and the structure of MAs can cause different host immune responses. Whether TDM in *Mtb* can achieve immune evasion by changing the structure of MAs may be a new research direction in the future.

TMM biosynthesis and its transport are essential for *Mtb*. In the cytoplasm, MAs are combined with trehalose in the form of TMM [41]. Once transferred to the periplasm, TMM provides MA for the synthesis of AGM and TDM under the action of enzymes [42]. TMM has been reported to share some similarities with TDM in terms of inflammatory potential, and TMM can also be recognized by Mincle to activate inflammatory responses [43].

Finally, recent research reports that TDB is a short-acyl-chain structural analogue of TDM [44,45]. TDB activates the MYD88 pathway by recognizing Mincle, which, in turn, promotes the production of ROS to promote the secretion of inflammatory factors. In addition, activation of the MYD88 pathway enhanced phosphorylation of PtdIns3K, STAT1, and promoted macrophage-induced autophagy to control the intracellular proliferation of *Mtb* [46]. Meanwhile, TDB activates the intracellular multiprotein complex called the NLRP3 inflammasome [47] and induces Th17 responses [48] (Figure 3A). Due to its good immune induction effect, TDB is now used as the active component of CAF01, a tuberculosis vaccine in phase I clinical trials.

#### 2.2.3. Glucose Monomycolate (GMM)

Upon infecting host cells, Mycobacterium adapts to the intracellular environment rich in glucose, leading to a preference for the production of GMM over TDM [49]. Nevertheless, the immunomodulatory role of GMM remains a subject of debate. Ishikawa and colleagues demonstrated that GMM exhibits weak recognition by the Mincle receptor, representing one of Mycobacterium’s strategies to evade host immunity [18]. Conversely, Hermann’s research identified Mincle receptor recognition of GMM [50]. This discrepancy in experimental outcomes may arise from Hermann’s utilization of a purified synthetic GMM product, in contrast to Ishikawa’s use of a product resulting from trehalase treatment. When considering both findings, it appears that GMM’s impact on innate immunity likely necessitates co-stimulation by other substances to manifest fully.

While the precise role of glucose monomycolate (GMM) in innate immunity remains somewhat elusive, substantial evidence highlights its proficiency in regulating adaptive immunity. Upon infection of antigen-presenting cells by *Mtb* lipid antigens, CD1 molecules are synthesized and can form complexes with these lipid antigens in the endoplasmic reticulum. Subsequently, these CD1–lipid antigen complexes translocate to the cell membrane surface, where they become recognizable by CD1-restricted T cells, thereby facilitating helper and effector functions. Our research demonstrates that T cell recognition of GMM relies on T cell receptors (TCRs) and exhibits remarkable specificity toward the native GMM structure produced by mycobacteria. This suggests that T cells possess the capability to intricately discern the hydrophilic cap of an antigen, potentially functioning as a classical TCR epitope by direct interaction with the TCR variable region [51,52]. In addition, GMM must be synthesized with glucose provided by mammals, which means that GMM production can be used as a good indicator of local invasion of mycobacteria, and its detection by the host immune system is helpful for effective monitoring of mycobacterial infection [53].

### 2.3. Immunomodulatory Effects of Non-Glycosylated Mycolic Acids of Mycobacteria

The glycosylated MAs mentioned above are mainly synthesized in actively replicating mycobacteria, while their levels are significantly lower in dormant mycobacteria [54]. Non-glycosylated MAs lipids such as glycerol monomycolate (GroMM) and free Mas are involved in biofilm formation and immune suppression. Accumulating evidence emphasizes the importance of non-glycosylated MAs in the invasion of *Mtb*. Therefore, understanding the immune modulation effects of non-glycosylated MAs is important for unraveling the immune evasion strategies of *Mtb* (Figure 3A).

#### 2.3.1. Glycerol Monomycolate (GroMM)

GroMM is selectively recognized by human Mincle but not mouse Mincle due to the lack of a sugar moiety [55]. The different reactivity of mice and humans to GroMM seems to explain the different responses activated by different hosts upon *Mtb* infection. Hence, more detailed analyses of human-specific Mincle may elucidate more detailed immune escape strategies for *Mtb*. In addition, GroMM also has been recognized by CD1b. GroMM-specific, CD1b-restricted T cells have been detected in the circulation of patients with latent, but not active tuberculosis. This seems to explain that during latent infection in humans, dormant *Mtb* preferentially produces GroMM [56].

#### 2.3.2. Free Mycolic Acids (fMAs)

The bioavailability of fMAs during natural infection has been a hot issue. Liposomes or other carriers are used as vehicles of water-insoluble MAs when studying the immune modulation of the host by fMAs. Since liposomes can effectively target phagocytic cells so that free mycolic acid is mainly located intracellularly, this may be the reason why transmembrane PRRs such as TLR2 and TLR4 reportedly do not activate inflammation [57]. However, recent studies have proven to disrupt this theory. Studies indicate that fMAs trigger receptors expressed on macrophage 2 (TREM2), which play an important role in the downstream signal induction. TREM2 is a surface receptor belonging to the family of TREM transmembrane glycoproteins. Studies have shown that the DNAX-activating protein12 (DAP12), an adaptor of TREM2, is consistently expressed in microglial cells (MC) [58,59]. TREM2 binding to DAP12 phosphorylates the immunoreceptor tyrosine-based activation motif (ITAM) in DAP12 and recruits Syk [60], activating phosphatidyl inositol protein kinase-3 (PI3K), and triggering the kinase cascade through PI3K/Akt signaling pathway to regulate the production, secretion, and release of inflammatory cytokines [61]. TREM2 is well characterized in neurodegenerative diseases. Overexpression of TREM2 can enhance the function of small MC phagocytic neurons, increase the gap between nerve debris and amyloid peptide in AD patients, and promote MC to clear amyloid precipitates [62]. However, a role for TREM2 in mediating inflammation and activation of macrophages has also emerged for *Mtb* infections. Ankita identified that infection of macrophages by *Mtb* is recognized by TREM2 and upregulates TREM2 expression through activation of the STING pathway, which, in turn, increases IL-10 and IFN-β expression. Increased levels of IFN-β are responsible for type I IFN-driven inhibition of ROS production and proinflammatory cytokine production, which caused intracellular survival of bacteria [63] (Figure 3). Further studies showed that the recognition of *Mtb* by TREM2/DAP12, but not CARD9, was dependent on fMAs. Furthermore, the knockdown of TREM2 resulted in a significantly enhanced Mincle-induced inflammatory response and facilitated mycobacterial clearance in vivo. This suggests that TREM2/DAP12 has an antagonistic effect on the Mincle–FcRγ–CARD9 pathway [64].

The composition of MA-containing lipids dynamically changes in response to the external environment. For example, the level of glycosylated MAs in *Mtb* decreases in the non-replicative dormant state and the fMA content increases. Hence, *Mtb* may promote chronic infection during latent infection through recognition of TREM2 receptors by fMAs.

In addition, macrophage polarization determines the fate of the bacteria within the cell. Macrophages are broadly classified into M1 and M2, which have distinct roles in the inflammatory response [65]. M1 macrophages are pro-inflammatory, while M2 macrophages are anti-inflammatory and functioning on tissue repair. TREM2–DAP12 signaling seemingly confers M2-like anti-inflammatory properties on macrophages. Thus, *Mtb* may regulate macrophage polarization through TREM2–DAP12 signaling.

The fMAs are chemically heterogeneous, further confounding their role in the host response to infection. MAs are divided into three main groups: α-MAs, methoxy-MAs, and keto-MAs. α-MAs have two cyclopropane rings, most of which are in the cis configuration. The methoxy- and keto-MAs have a cis or trans cyclopropane ring. The spatial structure of cyclopropane seems to influence the virulence of *Mycobacterium*. Beken et al. [66] used a single synthetic MA isomer to investigate the relationship between the structure of this virulence factor and its inflammatory function and found that α-MAs are relatively inert and do not activate the inflammatory response, but methoxy-MAs and keto-MAs activate a mild inflammatory response. In trans-cyclopropane orientation, the partial inflammatory response of methoxy disappeared and the pro-inflammatory nature of keto-MAs turned to an anti-inflammatory one. Thus, the different innate immune activities of MAs depend on the oxygenation class and the cis- and α-methyl-trans-cyclopropane chemical composition.

The regulation of immunity by fMAs is not only through the TREM2 receptor but also by affecting biofilm synthesis. In the laboratory, bacteria are typically grown and studied as dispersed, planktonic, pure cultures. However, in nature, the preferred lifestyle of micro-organisms is a surface-attached, multi-species microbial community called biofilm, which is physiologically and phenotypically distinct from bacteria grown in a free-swimming planktonic state. Biofilms are strongly implicated in chronicity and transmission of *Mtb* infections [67]. Sambandan et al. report that keto-MAs are essential for membrane growth, and mutants lacking or depleted of these MAs are unable to form membranes. Meanwhile, the Δ*mmaA4* strain, which cannot produce keto-MA, reduces *Mtb* resistance to rifampicin [68]. *Mtb* biofilms containing fMAs not only conferred strong viability to bacteria in vitro but were also described in infected organisms. The presence of this biofilm was found within primary granulomas in guinea pigs. In addition, these microbial colonies colonized an acellular rim adjacent to the edge of the mineralizing central necrotic core. Further studies revealed that the free mycolic acid in these biofilms was released through the breakdown of TDM [69]. These data suggest that free mycolic acid appears to be a discriminatory marker between latent and active infections. Also, this implies that *Mtb* adapts to changes in the microenvironment of the organism by changing the form in which MAs is present.

### 2.4. Regulation of Lipid Metabolism in Macrophages by Mycolic Acids

Excessive accumulation of intracellular cholesterol is considered to be a disturbance of lipid homeostasis. In normal conditions, cholesterol within macrophages is tightly regulated. It was found that alveolar macrophages from mice infected with Mtb showed increased expression of liver X receptors (LXRs). The increased expression of LXRs promotes accumulation of cholesterol in cells, which provides a source of nutrition for pathogens or acts as a nutrient stock during latent infections [70,71].

In addition to providing nutrition for Mtb, cholesterol also affects the host’s innate immune response. During Mtb infection, there is a close association between cholesterol and phagosomes. Cholesterol mediates TACO, an actin-binding protein, and binds to phagosomes containing live mycobacteria [72]. When the present cholesterol on the phagosome is depleted, the tight correspondence between the phagosome membrane and the surface of the mycobacterium is loosened and fusion with the lysosome occurs, leading to the destruction of the pathogen [73].

Thus, modulation of host cholesterol concentration by Mtb may be important in persistent infection [74] (Figure 3B).

As an important part of the Mtb regulator of cholesterol, MAs directly decide the fate of the bacteria in the cell. Next, we summarized a lot of evidence to prove that MAs control the accumulation of cholesterol.

#### 2.4.1. MAs Attract Cholesterol to Release into the Biofilm

Due to the property that cholesterol can diffuse along concentration gradients in biological membranes without the aid of specific transport mechanisms, it is an ideal carbon source for pathogens. It has been shown that fMAs from *Mtb* can attract cholesterol to release into the biofilm [69]. Brzostek et al. proved that the ability of *Mtb* to accumulate cholesterol close to its lipid-free zone of the cell wall was related to the extractable lipid components, such as free mycolic acid and TDM. This cholesterol accumulation decreases the permeability of the cell wall for the primary antituberculosis drug, rifampin, and partially masks the mycobacterial surface antigens [75]. If MAs are also able to attract large amounts of cholesterol in vivo, this is likely an important way for the pathogen to use host lipids to promote its proliferation.

#### 2.4.2. MAs Direct Host Macrophages Phenotype

Induction of granuloma formation is a signature after the invasion of *Mtb* into the organs. Although granulomas are essential for the protective role of the organism, they also provide a harbor for *Mycobacterium*. Macrophages are the key effector cells in granuloma formation, which can differentiate into vacuolated multinucleated giant cells (MGC) or lipid-containing foam cells [76]. It was found that *Mtb* escaped through foamy macrophages to the cell-free caseous-like centers of granulomas [77,78]. The formation of foamy macrophages has an important relationship with lipid droplets (LDs). LDs are widespread cell organelles dedicated to the storage of neutral lipids such as cholesteryl esters [79]. *Mtb* preferentially catabolizes cholesterol as a source of nutrition, and its acquisition through a unique *Mtb* input system.

In a pioneering paper, researchers demonstrated that different structures of MAs differed in directing host macrophages toward an expanded vesicular or lipid-containing foam phenotype. Following infection, macrophages engulf *Mtb*. Once in the host, *Mtb* is retained in the phagosome and secretes its cell wall components, mainly lipids and mycolic acids, for transporting from the phagosome to the cytoplasm. Various host factors are present in the cellular environment as sensors and effectors. The nuclear receptor TR4 was found to bind to secreted mycolic acid (keto-MAs), leading to downstream regulation of genes related to lipid biogenesis. CD36, a scavenger receptor known to be involved in lipid uptake, is regulated by TR4, leading to lipid droplet formation and increased foamy macrophage [80]. In contrast, methoxy-MAs were found to induce vesicle formation, a characteristic of the giant cells found in *Mtb* lung granulomas. Although cholesteryl ester content was unchanged, the ability to maintain and promote the growth of mycobacteria was increased [81]. However, Dubnau et al. used a *hma*-KO mutant *Mtb* H37Rv, which lacks all oxygenated MAs, and while the virulence of *Mtb* was significantly reduced, *Mtb* proliferation was still detectable in the lungs and spleen. This implies that MA is not a necessary substance to determine the survival of *Mtb*.

## 3. Conclusions

In the past decades, research related to MAs has received extensive attention, focusing on the following three main points.

(1)Development of novel anti-tuberculosis drugs: MAs represent a reservoir of new targets urgently needed to combat *Mtb* strains. Targeting MAs biosynthesis and other pathways is important for novel and promising drug candidates being developed. For example, through molecular docking techniques, targeted drugs can be designed to inhibit the transcription of the *cmaA2* or *mmaA4* genes, thereby greatly diminishing the pathogenicity of *Mycobacterium* and achieving therapeutic efficacy against tuberculosis;(2)Vaccine and adjuvant development: The effects of MAs on immunity are wide-ranging, and can mediate the development of innate immunity such as secretion of inflammatory factors by macrophages, neutrophils, or dendritic cells. It can also mediate the generation of adaptive immunity, e.g., effector cells and memory T cells respond rapidly upon re-encountering MAs. The modification of MAs is important for the development of vaccines as well as adjuvants. For example, TDB is a short-acyl-chain structural analogue of TDM, which is currently used as the active ingredient in CAF01, a TB vaccine undergoing phase I clinical trials (Table 1).(3)MAs may serve as a novel diagnostic marker: MAs appear to be a marker of *Mtb* adaptation to the external environment. For example, *Mtb* is converted from TDM to GMM upon cell invasion due to the presence of large amounts of glucose inside the cell. In addition, during the active phase, MAs in *Mtb* are present in the form of TDM, whereas during the dormant phase, MAs in *Mtb* exist in a free state. Thus, MAs may be included in studies of markers for detection of latent infection.

However, this does not mean that the whole research on MAs can be completely put into clinical application. Many issues remain to be resolved. Some of these issues are described below.

(1)The functions of many genes involved in the synthesis of MAs are still unknown, e.g., in *Mycobacterium*, all mycobacteria have two positions, i.e., distal and proximal, which initially contain a double bond that is subsequently modified to cis-cyclopropane, trans-double bond, cyclopropane, epoxide, or hydroxyl groups with adjacent methyl branches. The proteins that catalyze these desaturation/isomerization steps, as well as the underlying mechanisms, remain to be discovered and investigated;(2)Designing targeted drugs for a single gene may not result in good bacterial inhibition. For example, there are many genes that introduce methoxy or keto groups to MAs, and the absence of a single gene does not inhibit bacterial viability at all;(3)The inflammatory response induced by MAs has long been recognized as a double-edged sword, and how to rationally control the inflammation induced by MAs is a major problem to be solved in the future. Certainly, we cannot overlook the issue of MA-induced lipid accumulation in macrophages. The substantial lipid content within macrophages provides ample nutrition for *Mtb*. Balancing inflammation and lipid metabolism is also a challenge that needs to be addressed in the future;(4)Although *Mtb* changes the presence of MAs in different environments, it is too difficult to detect the pathogens in clinical practice. If the structure of MAs can be found to be associated with certain proteins in the host body, it seems that it can be an important diagnostic marker to distinguish between latent and active infections.

## Figures and Tables

**Figure 1 ijms-25-00396-f001:**
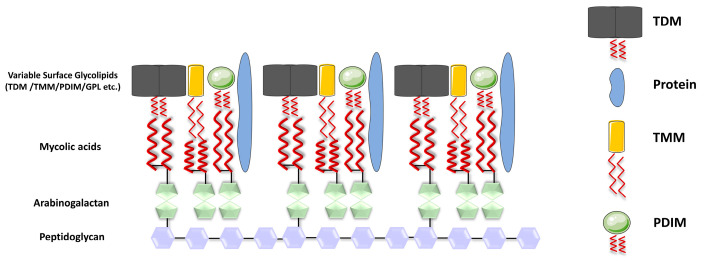
Structure of the cell wall of *Mycobacterium tuberculosis*. The cell wall of *Mtb* is formed by parallel arrangement of MA chains (represented by red wavy lines), which are linked to AG (depicted as green pentagons), while AG is covalently attached to peptidoglycan (PG) (shown as purple hexagons); the inner layer of the mycomembrane is presumably composed of free lipids, including TDM (depicted in gray), TMM (in yellow), and PDIM (in green spheres), along with the presence of proteins.

**Figure 2 ijms-25-00396-f002:**
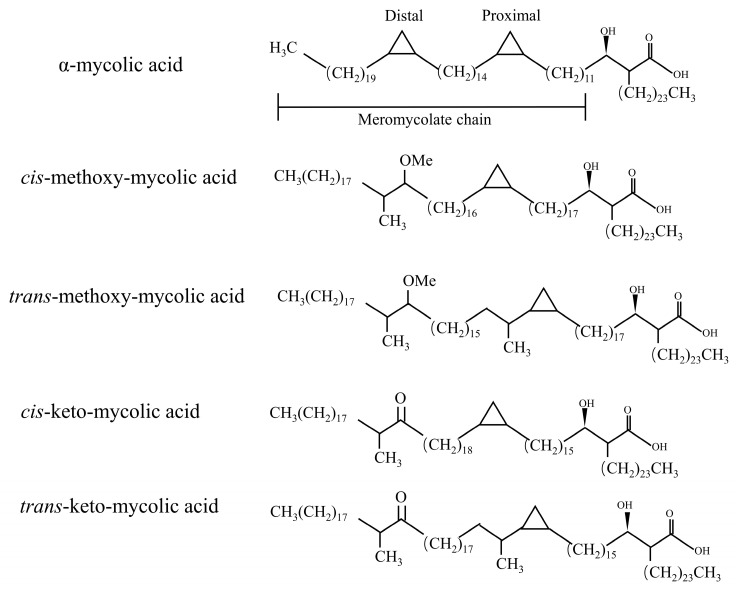
Major types of mycolic acid from *M. tuberculosis* complex. A α-MAs molecule has two cis-cyclopropane ring structures. The keto- and methoxy-mycocerosic acid molecules have one cis and one trans-cyclopropane ring structure.

**Figure 3 ijms-25-00396-f003:**
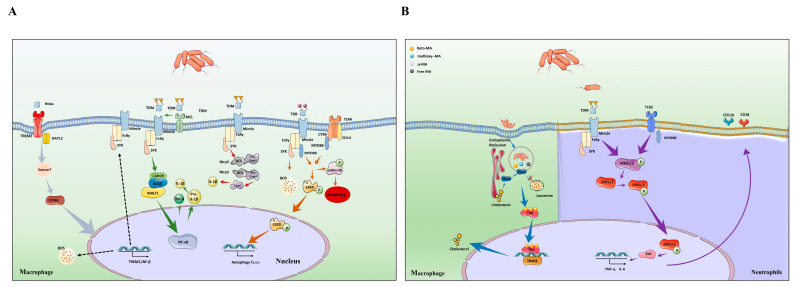
Different types of mycolic acids activate host cellular immunity and cholesterol accumulation. (**A**) TDM binding to Mincle triggers activation and upregulation of the NLRP3 inflammasome in macrophage, leading to release of caspase-1, which can promote the maturation of IL-1β (red arrow). In addition, TDM also promote the activation of Mincle–Syk-CARD9/Bcl10/MALT1 complex to mediate NF-κB signaling pathway to produce IL-1β (green arrow). MCL binds to Mincle to help recognize TDM. The synthetic analogue of TDM is TDB, which is an agonist for Mincle. TDB activates MYD88 pathway and enhances cytokine secretion through recognizing Mincle and TLR4 receptors. In addition, MYD88 pathway phosphorylates PtdIns3K, STAT1 to promote autophagy. This mechanism played a key role in controlling the growth of *Mtb* (orange arrow). Non-glycosylated mycolic acids play roles in host immunity. Free MAs are recognized by TREM2 receptor, which induces STING-dependent upregulation of TREM2 expression, which, in turn, enhances IFN-β and IL-10 secretion. The secretion of IFN-β inhibits proinflammatory cytokine production, which results in the increased intracellular survival of *Mtb*. Meanwhile, the expression of TREM2 inhibits the activation of Mincle to aggravate *Mtb* infection (gray arrow). (**B**) Mincle triggers phosphorylation of the MEK1/2 in neutrophils, resulting in the phosphorylation of ERK1/2 protein into the nucleus to promote the secretion of cytokines and the expression of CD11b and CD 18 molecules (purple arrow). (**B**) Once inside the host, *Mtb* is retained within the phagosomes, and secretes its cell wall components. Thereafter, the secreted components, mainly lipids and mycolic acid, are trafficked from the phagosome into the cytosol. Various host factors are present as sensors and effectors in the cellular milieu. The nuclear receptor TR4 binds to the secreted mycolic acid (keto-MA). Activation of the receptor by ligand binding leads to downstream regulation of gene involved in lipid biogenesis through its binding to its target gene having the TR4 response element (blue arrow).

**Table 1 ijms-25-00396-t001:** Different forms of mycolic acids and their immunomodulatory action.

Type of Mycolic Acids	Receptor	Immunomodulatory Action	Ref.
Trehalose dimycolate (TDM)	Mincle, MCL	1. Production of inflammatory cytokines; 2. Promotes neutrophil adhesion.	[23,24,31]
Trehalose dibehenate (TDB)	Mincle	1. Production of inflammatory cytokines; 2. Promoted macrophage induced autophagy	[41,43]
Glucose monomycolate (GMM)	CD-1	1. Activation of adaptive immune; 2. As a good indicator of local invasion of mycobacteria	[48,49,50]
Glycerol monomycolate (GroMM)	Mincle, CD-1	1. Production of inflammatory cytokines; 2. Distinguishing latent and active tuberculosis	[52,53]
Free mycolic acids (fMAs)	TREM2	1. Inhibition of inflammatory cytokines; 2. Induction of secretion INF-β	[60]

## Data Availability

The study did not produce any new or raw data.

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
