# Peer review of "Effect of Mycolic Acids on Host Immunity and Lipid Metabolism"

_ijms, 2023, doi:10.3390/ijms25010396_

Round 1

Reviewer 1 Report

This review article discussed the regulatory role of different mycolic acids and their immunomodulatory action and their role in metabolic remodeling especially in lipid metabolism, which play an important role in mycobacterial evasion of immune responses. As authors described, understanding mycolic acids-mediated immune response is very important to develop new drug targets as well as better treatment strategies against tuberculosis. Authors must address the following concerns:

1.       Authors should include a table summarizing the different forms of mycolic acids and their immunomodulatory action with clinical significance including citation of work.

2.       Line#70-72. Authors mentioned ‘the levels of these different species of MAs vary widely among mycobacterium species and strains, and numerous researchers have suggested that these differences in MAs content appear to be directly related to virulence.’ This portion should be discussed in detail by incorporating cell wall composition of different clinical strains as well as a- virulent strain of M. tuberculosis with respective MA and its contribution in virulence.

3.       Mycobacterial aggregation and cording are a very important factor in virulence of Mtb. Does different form MA play significant role bacterial aggregation? Should be discussed.

4.       Authors should discuss drug targeting of different MAs and its clinical significance. This section may describe the preclinical and clinical studies that target MAs synthesis and its functions.

5.       Line 71: Authors mentioned ‘numerous researchers have suggested that these differences in MAs content appear to be directly related to virulence’. Citations should be provided. I suggest authors to elaborate this section.

6.       Authors should include future prospective of their conclusion with future directions.

7.       Italicize the bacterial name throughout the manuscript.

8.       The quality of the figure should be increased.

9.       Line#404 is not clear.

10.   Line#405-412: what is the importance of this text in this manuscript? It is not clear.

Good

Author Response

Point by point response to the reviewer’s comments

Dear Editors and Reviewers:

Thank you for your letter and for the reviewers’ comments concerning our manuscript entitled “Effect of Mycolic Acids on Host Immunity and Lipid Metabo-lism” (ID: ijms-2594050). Those comments are all valuable and very helpful for revising and improving our paper, as well as the important guiding significance to our researches. We have studied comments carefully and have made correction which we hope meet with approval. Revised portion are marked in highlight yellow in the paper. The main corrections in the paper and the responds to the reviewer’s comments are as flowing:

Review1

  1. Authors should include a table summarizing the different forms of mycolic acids and their immunomodulatory action with clinical significance including citation of work.

Response: Thank you for your suggestions. We have added a table summarizing the different forms of mycolic acids and their immunomodulatory action

Type of mycolic acids

Receptor

immunomodulatory action

Ref.

trehalose dimycolate (TDM)

Mincle, MCL

1.Production of inflammatory cytokines

2. Promotes neutrophil adhesion.

[23],[24]

[31]

trehalose dibehenate (TDB)

Mincle

1.Production of inflammatory cytokines

2. Promotes macrophage induced autophagy

[41]

[43]

Glucose monomycolate (GMM)

CD-1

1.Activation of adaptive immune cells

2. As a good indicator of local invasion of mycobacteria

[48][49]

[50]

Glycerol monomycolate (GroMM)

Mincle, CD-1

1.Production of inflammatory cytokines

2.Distinguishing latent and active tuberculosis

[52]

[53]

Free mycolic acids (fMAs)

TREM2

1.Inhibition of inflammatory cytokines

2.Induction of secretion INF-β

[60]

  1. Line#70-72. Authors mentioned ‘the levels of these different species of MAs vary widely among mycobacterium species and strains, and numerous researchers have suggested that these differences in MAs content appear to be directly related to virulence.’ This portion should be discussed in detail by incorporating cell wall composition of different clinical strains as well as a- virulent strain of tuberculosis with respective MA and its contribution in virulence.

Response: Thank you for your suggestions. We have elaborated contents with comparison the variations of different types of MAs between various mycobacterial species and strains, citing examples such as non-tuberculous mycobacteria and BCG (Line#85-96).

  1. Mycobacterial aggregation and cording are a very important factor in virulence of Mtb. Does different form MA play significant role bacterial aggregation? Should be discussed.

Response: Thank you for your suggestions. TDM has been identified to be the cord factor. We have explored the impact of mycolic acid structural variations on trehalose dimycolate (TDM), indirectly substantiating the influence of mycolic acids on bacterial virulence. (Line#214-230)

  1. Authors should discuss drug targeting of different MAs and its clinical significance. This section may describe the preclinical and clinical studies that target MAs synthesis and its functions.

Response: Thank you for your suggestions. We have described the process of MAs synthesis in detail (Line#49-73). This process involves a number of enzymes, which have the potential to be targets for anti-tuberculosis drug development, as we have exemplified (Line#99-106).

  1. Line 71: Authors mentioned ‘numerous researchers have suggested that these differences in MAs content appear to be directly related to virulence’. Citations should be provided. I suggest authors to elaborate this section.

Response: Thank you for your suggestions. We have elaborated on the significant variations in the levels of different types of MAs between various mycobacterial species and strains, citing examples such as non-tuberculous mycobacteria (Line#85-96).

  1. Authors should include future prospective of their conclusion with future directions Response: Thank you for your suggestions. We have re-written the conclusion section. We have summarized the entire paper and highlighted the current research challenges and future prospects regarding MAs (Line#420-463)
  2. Italicize the bacterial name throughout the manuscript.

Response: We have made correction according to the Reviewer’s comments. We have made corrections to the corresponding errors in the text.

  1. The quality of the figure should be increased.
  2. Response: Thank you for your suggestions. We have made modifications to the figure.
  3. Line#404 is not clear.

Response: Thank you for your suggestions. We have re-written the conclusion section.

Line#405-412: what is the importance of this text in this manuscript? It is not clear. Response: Thank you for your suggestions. We have re-written the conclusion section.

Please refer to the attachment for the revised manuscript

Reviewer 2 Report

In this review article, Wang et al. describe the emerging connections between free and attached mycolic acids and the activation of host immunity and lipid metabolism. After initially defining and generally summarizing mycolic acid structure and assembly, the authors delve into detail about diverse signaling pathways and the connection between mycolic acids and host immunity. The breakdown of the subsections based on varying mycolic acid conjugates and free mycolic acids sets a clear article structure for understanding their divergent immunological impacts. The main figure (Figure 3) provides a comprehensive graphical representation of the written description but is too small and condensed to fully support the written descriptions. A couple of descriptions need expansion and the entire document needs a careful edit, see detailed comments below, but with these minor revisions, this review should be ready for publication in IJMS.

1.     Figures

a.     Figure 1: This figure needs a more complete figure legend to explain the coloration of the variable attachments on the mycolic acids and the variable structures of the mycolic acid chains. Incorporating some graphical representation of the MAs assembly process would also better support the written description in 2.1 (see below).

b.     Figure 3: This figure fully encapsulates all of the pathways and details described in the rest of the article. However, the amount of information contained in the figure is impossible to digest in its current state. The figure either needs to fill an entire page to be useable in its current form, which will require rearranging the layout of the pathways to fit clearly onto an entire page, or preferably this figure needs to be broken into multiple smaller subfigures that are placed alongside the corresponding written subsections. After an overview figure, this figure could be broken into the green subsection (left side), the light green subsection (middle), and blue subsection (right side) to make it easier to differentiate these various pathways and their cellular locations (macrophages, neutrophils, and overview). Time put into this figure will significantly help the reader digest all of the written descriptions to keep each of the complex signaling pathways straight.

2.     A couple of key sections in the review need revision to clarify the material presented.

a.     In the introductory section to MAs (2.1), the authors clearly wanted to keep this description brief but the brevity of the current description makes it impossible to follow the information. This entire section needs expansion and careful revision, but a few key challenges are highlighted below.

                                               i.     The start of this section needs some context. Where is the synthesis of MAs occurring? When does this occur? Where does the starting material for this synthesis come from?

                                             ii.     The first paragraph tries to cover too much information too quickly. For example, the first sentence covers 5 lines with 5/6 interconnected clause statements and tries to go from MA construction to the multiple enzymes catalyzing this condensation reaction in a single sentence. Each of these components should have their own sentences and likely added sentences to introduce each component prior to putting them together into this overall pathway.

                                            iii.     In paragraph 2, the chemical attachment between MAs and sugar/peptidoglycan is not clear. The authors need to more carefully and, in more detail, describe the chemical attachments between MAs.

b.     Conclusion paragraph. The conclusion paragraph lacks a clear structure, which leads the authors to ramble through a series of disconnected sentences rehashing some details from the article and interjecting some new information. I think that this conclusion section would be best organized into two paragraphs where the first paragraph summarizes the key points from the review and the second paragraph points to gaps of knowledge and areas for future study.

                                               i.     Lines 381-384: This reviewer read this sentence multiple times but still cannot make sense of what the authors are trying to say.

1. The article needs to be carefully edited for proper word choice and sentence structure. The lack of careful edits is especially glaring because multiple default sentences and statements are still present from the article template (see conclusions section for example vestigial template statements.)

a.     Also see subsection headings 2.2.1 and 2.2.3 for necessary edits.

b.     Formatting of MA/Ma switches throughout the article and should be standardized.

c.     Scientific names should be italicized.

Author Response

Point by point response to the reviewer’s comments

Dear Editors and Reviewers:

Thank you for your letter and for the reviewers’ comments concerning our manuscript entitled “Effect of Mycolic Acids on Host Immunity and Lipid Metabo-lism” (ID: ijms-2594050). Those comments are all valuable and very helpful for revising and improving our paper, as well as the important guiding significance to our researches. We have studied comments carefully and have made correction which we hope meet with approval. Revised portion are marked in highlight yellow in the paper. The main corrections in the paper and the responds to the reviewer’s comments are as flowing:

Review 2

  1. a.     Figure 1: This figure needs a more complete figure legend to explain the coloration of the variable attachments on the mycolic acids and the variable structures of the mycolic acid chains. Incorporating some graphical representation of the MAs assembly process would also better support the written description in 2.1 (see below).

Response: Thank you for your suggestions. We have added the figure legend as per your suggestions. (Line#113-117)

  1. Figure 3: This figure fully encapsulates all of the pathways and details described in the rest of the article. However, the amount of information contained in the figure is impossible to digest in its current state. The figure either needs to fill an entire page to be useable in its current form, which will require rearranging the layout of the pathways to fit clearly onto an entire page, or preferably this figure needs to be broken into multiple smaller subfigures that are placed alongside the corresponding written subsections. After an overview figure, this figure could be broken into the green subsection (left side), the light green subsection (middle), and blue subsection (right side) to make it easier to differentiate these various pathways and their cellular locations (macrophages, neutrophils, and overview). Time put into this figure will significantly help the reader digest all of the written descriptions to keep each of the complex signaling pathways straight.

Response: Thank you for your suggestions. We split the figure into two parts to make it easier for readers to read and added the figure legend.

  1.   A couple of key sections in the review need revision to clarify the material presented.
  2. In the introductory section to MAs (2.1), the authors clearly wanted to keep this description brief but the brevity of the current description makes it impossible to follow the information. This entire section needs expansion and careful revision, but a few key challenges are highlighted below.
  3. The start of this section needs some context. Where is the synthesis of MAs occurring? When does this occur? Where does the starting material for this synthesis come from?
  4. The first paragraph tries to cover too much information too quickly. For example, the first sentence covers 5 lines with 5/6 interconnected clause statements and tries to go from MA construction to the multiple enzymes catalyzing this condensation reaction in a single sentence. Each of these components should have their own sentences and likely added sentences to introduce each component prior to putting them together into this overall pathway. 第

iii.  In paragraph 2, the chemical attachment between MAs and sugar/peptidoglycan is not clear. The authors need to more carefully and, in more detail, describe the chemical attachments between

Response: Thank you for your excellent advice on the revision of the introduction part. We have described the process of MAs synthesis in detail (Line#49-73), and we have recharacterized the way in which MA is linked to peptidoglycan (Line#62-73).

Conclusion paragraph. The conclusion paragraph lacks a clear structure, which leads the authors to ramble through a series of disconnected sentences rehashing some details from the article and interjecting some new information. I think that this conclusion section would be best organized into two paragraphs where the first paragraph summarizes the key points from the review and the second paragraph points to gaps of knowledge and areas for future study. Lines 381-384: This reviewer read this sentence multiple times but still cannot make sense of what the authors are trying to say.

Response: Thank you for your suggestions. We have re-written the conclusion section which inclued two paragraphs where the first paragraph summarizes the key points from the review, and the second paragraphs points to gaps of knowledge and areas for future study (Lin#420-463).

The article needs to be carefully edited for proper word choice and sentence structure. The lack of careful edits is especially glaring because multiple default sentences and statements are still present from the article template (see conclusions section for example vestigial template statements.)

  1. Also see subsection headings 2.2.1 and 2.2.3 for necessary edits.

Response: Thank you for your suggestions. We have modified the statement in Line#151-161 and Line#254-278

  1. Formatting of MA/Ma switches throughout the article and should be standardized.

Response: Thank you for your suggestions. We have modified the spelling mistake.

  1. Scientific names should be italicized

Response: Thank you for your suggestions. We have modified all the scientific names in italics.

Finally, we tried our best to improve the manuscript and made some changes in the manuscript.  These changes will not influence the content and framework of the paper. And here we did not list the changes but marked in highlight yellow in revised paper.

We appreciate for Editors/Reviewers’ warm work earnestly, and hope that the correction will meet with approval.

Once again, thank you very much for your comments and suggestions.

Prof. Xiangmei Zhou

China Agricultural University

Beijing, China

Tel: +86-010-62734618.
